# Outpatient Microdose Induction with Transdermal Buprenorphine: A Case Series

**DOI:** 10.3390/healthcare10071307

**Published:** 2022-07-14

**Authors:** Shannon Menard, Archana Jhawar

**Affiliations:** 1Department of Pharmacy Services, Jesse Brown VA Medical Center, 820 South Damen Ave, Chicago, IL 60612, USA; archanaj@uic.edu; 2College of Pharmacy, Department of Pharmacy Practice, University of Illinois at Chicago, 833 South Wood St, Chicago, IL 60612, USA

**Keywords:** drug addiction, opioids, buprenorphine, methadone, microdose

## Abstract

Transdermal buprenorphine is FDA approved for chronic severe pain but has an increasing amount of data supporting its use to transition patients from full opioid agonists to sublingual buprenorphine via a microdose strategy. The literature has primarily focused on patients with a pain diagnosis or who have been prescribed opioids in inpatient units. This case series reviews the use of transdermal buprenorphine to transition patients from methadone and illicit opioids to sublingual buprenorphine. The authors identified seven patients from an outpatient opiate treatment program who received the transdermal buprenorphine protocol. All patients were prescribed methadone and used illicit heroin prior to and during the transition. Five patients (71.4%) successfully completed the transition to sublingual buprenorphine, with all five patients reporting no or mild withdrawal symptoms. These findings suggest that transdermal buprenorphine is a potentially safe and effective microdose induction method for patients who use illicit substances in an outpatient setting.

## 1. Introduction

Buprenorphine is a mu-opioid receptor partial agonist approved for the treatment of opioid dependence. It has a strong affinity to the mu receptor, allowing it to displace full agonists and to precipitate withdrawal symptoms [1,2]. To avoid this, two methods have been successfully implemented. Traditionally, buprenorphine is initiated once a patient begins to experience withdrawal symptoms, but this can result in delays in treatment initiation, reduced patient retention, and increased relapses [1,3,4,5]. The other method is initiating microdoses of buprenorphine while the patient is still using a full opioid agonist [6,7,8]. This method allows the buprenorphine to gradually replace the full agonist in the mu receptors, titrate to a therapeutic dose, and prevent withdrawal symptoms until the full agonist can be abruptly discontinued. This strategy has primarily been shown with the combination sublingual (SL) buprenorphine/naloxone products.

Currently, the transdermal buprenorphine monoproduct is not FDA approved for the treatment of opioid use disorder (OUD), but there are published case reports and case series utilizing this formulation to slowly transition patients via a microdose strategy. Transdermal buprenorphine delivers small doses of 5 to 20 mcg per hour, which is less than 0.5 mg/day of buprenorphine [9]. This was first introduced in 2011, when researchers successfully transitioned 10 of 11 patients (91%) from 60–100 mg of methadone to SL buprenorphine [10]. A 35 mcg patch was applied 12 h after the last dose of methadone, and SL buprenorphine was initiated on day two. The SL dose was titrated to 8 mg by day four, when the patch was removed. There were no adverse events reported during this transition, with the average withdrawal symptoms categorized as mild. Since this case series, there have been several reports utilizing transdermal buprenorphine as a microdose strategy; however, these have primarily been in inpatient settings for patients not using illicit opioids or who are using prescribed opioids for pain management [11,12,13,14]. An inpatient setting allows for a controlled environment, frequent monitoring, and easily adjustable doses [11]. The focus on the inpatient use of microdose transitions inadvertently results in a large cohort of patients lacking data on safety and efficacy in the outpatient setting.

There are limited outpatient data exemplifying a safe microdose transition via transdermal buprenorphine. Studies often included a mixed group of patients using prescribed opioids for pain, illicit opioids, or those transitioning from methadone [12,15,16]. However, the authors were unable to find literature supporting the outpatient use of transdermal buprenorphine to transition patients using illicit opioids via a microdose strategy in the general patient population. In one study, eight pregnant patients (four outpatients and four inpatients) were transitioned from illicit fentanyl +/− morphine or tramadol to SL buprenorphine utilizing two 10 mcg patches [15]. A 2 mg dose of SL buprenorphine was initiated 24 h after patch application and titrated to 16 mg over two days, with patch removal at 48 h. All of the patients reported no or mild withdrawal symptoms. To the authors’ knowledge at the time of submission, this is the first published study in the literature outlining a safe transition with the use of transdermal buprenorphine from illicit opiates via a microdose strategy in the general outpatient population. All of the patients described below were enrolled in an opiate treatment program, receiving methadone, and using additional illicit opioids during the time of transition from 1 January 2016 to 31 December 2021.

## 2. Cases

### 2.1. Case 1

A 68-year-old black male was prescribed 80 mg of methadone daily and used 3–4 bags of heroin every two days via insufflation. A urine drug screen at the time of microdose initiation was positive for benzodiazepines, cocaine, opiates, and methadone. Other documented concurrent substance use included a six-pack of beer 4–5 times a week. On day 1, one transdermal 20 mcg buprenorphine patch was applied, and 80 mg of methadone was continued. The second patch was to be applied on day 4, but the patient was not present at his follow-up appointment until the following day. On day 5, a second 20 mcg patch was applied while continuing 80 mg of methadone. There were no reported withdrawal symptoms in the nursing or psychiatry notes at this point in the transition. On day 12, the last dose of methadone was administered. On day 13, the transdermal patches were removed, and 2 mg/0.5 mg of SL buprenorphine/naloxone was administered under direct supervision. The patient was monitored for over an hour without any noted withdrawal symptoms. He successfully transitioned without any documented withdrawal and was to return on day 14 for the continued titration of SL buprenorphine/naloxone; however, the patient was not present at any appointments and did not respond to outreach and was discharged from the clinic.

### 2.2. Case 2

A 63-year-old black male was prescribed 50 mg of methadone daily and was documented as using one bag of heroin both “every couple of days” and “a couple of times a week” via insufflation. A urine drug screen was positive for opiates and methadone at the time of microdose initiation. Due to prolonged QTc of 507 milliseconds, methadone dose was decreased to 40 mg four days prior to transdermal buprenorphine initiation. On day 1, 20 mcg of transdermal buprenorphine was applied and 40 mg of methadone was continued. On day 2, a second 20 mcg patch was applied without any opioid withdrawal symptoms documented. The patient was to return to the clinic on day 9, but instead arrived on day 10. Both transdermal patches were removed, methadone was discontinued, and 2 mg/0.5 mg of SL buprenorphine/naloxone was administered with monitoring. The patient denied any withdrawal symptoms or side effects during the transition. The patient was to return on day 11 for buprenorphine/naloxone titration; however, he did not arrive for appointments or respond to outreach until day 19, when he reported opioid cravings and last heroin use on day 14. There were no opioid withdrawal symptoms documented. Buprenorphine/naloxone was increased to 4 mg/1 mg SL daily. On day 22, the dose was further titrated to 8 mg/2 mg with ongoing cravings but no withdrawal symptoms. Throughout the duration of transition, titration, and attempted maintenance, urine drug screens were persistently positive for opiates. During the microdose titration and low doses of treatment, the patient denied any withdrawal symptoms. However, as the buprenorphine/naloxone was titrated, he experienced withdrawal symptoms with concurrent illicit opiate use.

### 2.3. Case 3

A 70-year-old black male was prescribed 130 mg of methadone daily and used 1–2 bags of heroin “several days out of the week”. At the time of microdose initiation, urine drug screens were positive for benzodiazepines, opiates, and methadone. First, the methadone dose was reduced by 10 mg every two days until buprenorphine initiation began due to a prolonged QTc of 525 milliseconds. On day 1, 20 mcg of transdermal buprenorphine was applied, and 100 mg of methadone was continued. On day 2, a second 20 mcg patch was applied, and 100 mg of methadone was continued. At this time, the patient denied opioid withdrawal and was documented as saying “I feel perfectly normal”. On day 7, the patient continued to deny withdrawal symptoms, and methadone was further decreased to 90 mg. On day 9, methadone was discontinued, transdermal patches remained, and 2 mg/0.5 mg of SL buprenorphine/naloxone was administered with a 90 min monitoring period. After approximately one hour, the patient was documented as having “mild opioid withdrawal” and a short prescription for hydroxyzine 25 mg every six hours as needed was provided. On day 10, the patient reported withdrawal symptoms that only lasted 2 h and were then resolved without any further symptomatology. On this day, both transdermal patches were removed, and SL buprenorphine/naloxone was titrated to 4 mg/1 mg with plans to monitor; however, the patient left after receiving his dose. On day 13, the patient denied any withdrawal symptoms, cravings, or side effects from SL buprenorphine/naloxone, and the dose was increased to 8 mg/2 mg. On day 15, the dose was titrated further to 12 mg/3 mg, with no reported withdrawal symptoms and the first negative urine opioid screen in 9 months.

### 2.4. Case 4

A 58-year-old black male was prescribed 50 mg of methadone and used USD 20 of heroin daily via insufflation. Urine drug screens were positive for cannabinoids, cocaine, opiates, and methadone. The methadone dose was reduced to 40 mg prior to microdose induction for unknown reasons. On day 1, 20 mcg of transdermal buprenorphine was applied, and 40 mg of methadone was continued. The plan was to follow up on day 2 for the application of the second patch; however, documentation was not available for administration on day 2 or symptoms after administration. On day 8, the patient denied any opioid withdrawal symptoms, and two transdermal buprenorphine patches were documented as being removed; methadone was discontinued, and 2 mg/0.5 mg of SL buprenorphine/naloxone was administered. At this time, the urine drug screen was negative for opiates. On day 9, the patient reported “minor sweating” but was agreeable to dose titration and received 4 mg/1 mg. On day 10, he continued to endorse “slight sweating” but remained agreeable to further dose titration to 8 mg/2 mg and denied any side effects. On day 14, the patient reported feeling “general tiredness” that he attributed to “low testosterone” but denied any traditional withdrawal symptoms. His dose was continued, and urine drug screens remained negative.

### 2.5. Case 5

A 64-year-old black male was prescribed 60 mg of methadone daily and used one bag of heroin every 3–4 days via insufflation. At the time of microdose transition, a urine drug screen was positive for opiates and methadone and intermittently positive for cocaine. The patient also received a prescription for acetaminophen/codeine for headaches that he took two times a week. On day 1, 20 mcg of transdermal buprenorphine was applied, and 60 mg of methadone was continued. On day 2, the patient denied any withdrawal symptoms, and a second patch was applied, and 60 mg of methadone was continued. On day 5, the patient denied any withdrawal symptoms during the transition. On day 8, the patient continued to deny side effects or withdrawal symptoms. Both transdermal patches were removed, methadone was discontinued, and 2 mg/0.5 mg of SL buprenorphine/naloxone was administered with 1 h monitoring. On day 10, the patient continued to deny withdrawal symptoms and opioid cravings, and the dose was increased to 8 mg/2 mg. On day 12, the primary care provider noted that the patient was asking for acetaminophen/codeine to manage headaches, but the provider refused to prescribe. He continued with treatment after a successful transition to buprenorphine/naloxone but continued to use illicit opiates.

### 2.6. Case 6

A 63-year-old black male was prescribed 150 mg of methadone daily and used three bags of heroin daily via insufflation. A urine drug screen was positive for opiates and methadone at the time of microdose induction. Nineteen days prior to transdermal application, the patient’s methadone dose was tapered by 20 mg daily to 80 mg daily. The patient denied withdrawal symptoms but reported increasing heroin use to avoid symptoms. On day 1, 20 mcg of transdermal buprenorphine was applied, and 80 mg of methadone was continued. On day 2, the patient denied withdrawal symptoms, and a second patch was applied, with continued 80 mg of methadone. On day 4, the patient documented having “mild withdrawal symptoms” but agreed to continue treatment. On day 5, the patient denied any withdrawal symptoms and had abstained from heroin for two days, with the most recent use consisting of only one bag. Both patches were removed at this time, methadone was discontinued, and 2 mg/0.5 mg of SL buprenorphine/naloxone was administered with 1 h of monitoring. The patient continued to deny side effects, and the plan was to return the following day for titration. On day 11, the patient presented to the clinic intoxicated, with heroin use that morning. He reported withdrawal symptoms from buprenorphine/naloxone products and requested the re-instatement of methadone. Descriptions such as severity and the symptoms of withdrawal were not documented.

### 2.7. Case 7

A 68-year-old black male was prescribed 60 mg of methadone and used one bag of heroin 2–3 times a week via insufflation. Urine drug screens were positive for opioids and methadone at the time of microdose initiation. Of note, the patient was resistant to transitioning to buprenorphine products at first but did so at the encouragement of the opiate treatment program due to several medical conditions that would limit methadone titration, including QTc prolongation, obstructive sleep apnea, and congestive heart failure. On day 1, the patient reported withdrawal symptoms of back pain, rhinorrhea, and loss of appetite due to not receiving his methadone the day prior to patch administration. A 20 mcg dose of methadone was applied, and 60 mg of methadone was continued. On day 2, the patient reported that the patch did not adhere properly and fell off; the patient was asked to return the following day for patch reapplication. On day 3, the 20 mcg patch was applied with overlaying adhesive pads with plans to return on day 6 for the second patch. The patient arrived at the clinic one day late on day 7, and the second patch was applied with overlaying adhesive pads, and 60 mg of methadone was continued. The patient reported minimal withdrawal symptoms of rhinorrhea and yawning. On day 9, he denied withdrawal symptoms but could not return until day 14 to transition to SL buprenorphine/naloxone. On day 14, the patient arrived but was reported to have COVID-19 exposure. With full personal protective equipment, the patches were removed, methadone was discontinued, a 2 mg/0.5 mg dose of SL buprenorphine/naloxone was administered, and 15 days of medication was provided. On day 34, the patient presented with moderate to severe withdrawal symptoms documented as anxiety, agitation, excessive irritability, rhinorrhea, lacrimation, nausea, vomiting, abdominal cramps, and severe diarrhea. He had stopped taking buprenorphine/naloxone on day 29 due to continued withdrawal symptoms even when taking a 4 mg/1 mg dose and began using heroin to mitigate withdrawal. He agreed to trial a 2 mg/0.5 mg dose of SL buprenorphine/naloxone and noted to have reduced withdrawal symptoms after 2 h and was agreeable to a second dose. He did not stay for monitoring and left after 20 min. On day 35, the patient did not show up to his appointment for titration. On day 37, the patient returned to the clinic with opiate withdrawal symptoms documented as excessive irritability, body aches, severe rhinorrhea and lacrimation, anxiety, agitation, anorexia, severe nausea, and insomnia. A 4 mg/1 mg dose of buprenorphine/naloxone was administered, and 2 h later, another 4 mg/1 mg dose was administered. On day 38, symptoms of lacrimation, rhinorrhea, and nausea continued, and the dose was increased to 12 mg/3 mg. Unfortunately, the patient tested positive for COVID-19, and the next note on day 56 documented continued lacrimation, rhinorrhea, and nausea; the dose was increased again to 16 mg/4 mg. The patient continued to experience withdrawal symptoms and had ongoing heroin use despite further dose titrations.

## 3. Discussion

In this case series of patients treated in an opiate treatment program and using varying amounts of illicit opioids, cases 1–5 (71.4%) were able to successfully transition from methadone 50–130 mg and varying amounts of heroin to buprenorphine/naloxone utilizing buprenorphine patches. Of these patients, cases 1, 2, and 5 (60%) denied withdrawal symptoms during the transition. The withdrawal symptoms experienced in cases 3 and 4 were described as mild and resolved quickly. The two patients who were unable to successfully transition, cases 6 and 7, had poor follow up within the opiate treatment program, which could have contributed to the poor tolerability of the microdose protocol. 

The method of transdermal buprenorphine microdose transition was patient dependent (See Table 1). All of the patients received two 20 mcg patches, with most patients having the second patch applied on the second day. Two patients, cases 1 and 7, had the second patch applied four days after the first. Methadone was continued during transdermal buprenorphine therapy. On average, the transdermal patches were removed 7 days after the second patch was applied. On the day patches were removed, methadone was discontinued without taper, and SL buprenorphine/naloxone 2 mg/1 mg was initiated. However, case 3 started SL buprenorphine/naloxone without the removal of the transdermal patches, experienced withdrawal symptoms within the monitoring window, and required prescribed hydroxyzine. This may demonstrate the importance of removing the patches when starting SL therapy. Despite this, the patient successfully transitioned.

The literature evaluating transdermal buprenorphine microdose inductions varies greatly regarding patient populations and dosing strategies. There are cases describing transitions in patients who use methadone or illicit substances [10,12,13,15,16,17,18]. However, patients who are receiving methadone for the treatment of OUD may continue to use illicit opioids throughout their treatment, which can complicate the transition to SL buprenorphine. The case series establishing successful microdose transition with transdermal buprenorphine from illicit opioids was performed in a pregnant patient population [15]. The present case series demonstrates that patients who use illicit heroin in varying amounts can successfully transition from methadone to SL buprenorphine/naloxone with the transdermal microdose approach in an all-black male population. Additionally, where most case series were performed in the inpatient population, this case series adds to the growing literature that suggests that transdermal buprenorphine may be a safe and effective method to transition patients from opioids to SL buprenorphine in an outpatient setting.

Compared to other reports in the literature, the cases in this report utilized a higher dose of buprenorphine patches, 40 mcg compared to 5–35 mcg. Many cases removed the patch after approximately 48 h and initiated SL buprenorphine/naloxone within two days [11,14,15,17]. By applying the patches for a longer period of time prior to initiating SL buprenorphine/naloxone, as seen in this case series, buprenorphine may have had more time to slowly replace the full opioid agonists on the mu-opioid receptor to smooth the transition to SL therapy.

This case series is the first to demonstrate novel microdose induction utilizing transdermal buprenorphine in a general outpatient population who used illicit substances at the time of induction. Though this was a retrospective chart review, which can limit the results to what is documented in the medical chart, most of the information sought in this review was available. There was also sufficient follow up recorded for most patients. However, this study does not come without limitations. The protocol varied with each patient and differed slightly from protocols in the literature, which makes it difficult to find a standard induction approach using buprenorphine patches. There were also no consistent trends across the cases that predicted which patients would be successful; however, the two unsuccessful cases were noted to have poor follow up during the transition.

## 4. Conclusions

Transdermal buprenorphine is a potentially safe and effective microdose induction method to transition patients from methadone and insufflated heroin to SL buprenorphine/naloxone, as demonstrated by this case series. This disorder and method of transition are notoriously challenging for providers and patients to manage. By researching safe alternative induction strategies, it allows providers multiple ways to engage patients in treatment. In addition to traditional induction, microdose approaches using SL or transdermal buprenorphine may be considered to improve induction experiences.

## Figures and Tables

**Table 1 healthcare-10-01307-t001:** Case summaries.

	Case 1	Case 2	Case 3	Case 4	Case 5	Case 6	Case 7
Methadone Dose	80 mg	50 mg tapered to 40 mg	130 mg tapered to 100 mg	50 mg	60 mg	150 mg tapered to 80 mg	60 mg
Heroin Amount Used	3–4 bags every 2 days	1 bag “couple times a week”	1–2 bags several days a week	USD 20 daily	1 bag every 3–4 days	3 bags daily	1 bag 2–3 times/week
First Buprenorphine Transdermal Patch	Day 1	Day 1	Day 1	Day 1	Day 1	Day 1	Day 1—fell off Day 3—reapplied
Second Buprenorphine Transdermal Patch	Day 5	Day 2	Day 2	Unknown	Day 2	Day 2	Day 7
Methadone Discontinued	Day 12	Day 10	Day 9	Day 8	Day 8	Day 5	Day 14
Buprenorphine Patches Removed	Day 13	Day 10	Day 10	Day 8	Day 8	Day 5	Day 14
Sublingual Buprenorphine/Naloxone Started	Day 13	Day 10	Day 9	Day 8	Day 8	Day 5	Day 14
Withdrawal Symptoms	None documented	None documented during transition ^1^	Day 9	Day 9, Day 10	None documented	Day 4, Day 11	Day 7 (mild), Day 34 (mod-severe)
Successful Transition	Yes	Yes	Yes	Yes	Yes	No	No

^1^ Case 2 experienced withdrawal symptoms during titration of sublingual buprenorphine after transition.

## Data Availability

Not applicable.

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
