# Peer review of "Outpatient Microdose Induction with Transdermal Buprenorphine: A Case Series"

_healthcare, 2022, doi:10.3390/healthcare10071307_

Round 1
Reviewer 1 Report
I read this case series with great interest. This is a practical and focused paper, and while I agree with the authors that they have shown that transdermal buprenorphine micro-dosing can work, it is also clear that this is a hard-to-reach patient group. In many places in Europe and in Canada, this group has been reached and kept in treatment with daily onsite diacetyl-morphine in addition to methadone ("heroin-assisted treatment"). I will leave it to the authors' discretion if they want to mention this type of treatment as an option.
Author Response
Thank you for the thoughtful response as this is indeed a hard-to-reach population. Though diacetyl-morphine has been successfully used in Europe/Canada, we have chosen to omit this from the report as the focus was on the use of transdermal buprenorphine as a micro-dose induction technique.
Reviewer 2 Report
Thank you for giving me the opportunity to read and comment a report “Outpatient microdose induction with transdermal buprenorphine: A case series”, by S. Menard and A. Jhawar.
This paper is well written, correctly structured with a suitable research concept, and it is of relevance to readers of the journal. However, I have a few comments to make below.
· According to the CARE (CAse REport) guideline, the "Discussion" section should include the strengths and limitations. In addition, the importance of each limitation should be highlighted.
· In the "Conclusions" section, a justification should be included, as well as the main clinical lessons learned. Therefore, the last sentence should be deleted or removed to the "Discussion" section.
· According to the CARE guideline, if possible, the patient's perspective or experience should be included.
Author Response
Thank you for giving me the opportunity to read and comment a report “Outpatient microdose induction with transdermal buprenorphine: A case series”, by S. Menard and A. Jhawar.
This paper is well written, correctly structured with a suitable research concept, and it is of relevance to readers of the journal. However, I have a few comments to make below.
Thank you for your thoughtful response, please see our responses in blue.
- According to the CARE (CAse REport) guideline, the "Discussion" section should include the strengths and limitations. In addition, the importance of each limitation should be highlighted.
We have added strengths and limitations to the manuscript, please see Lines 252-261.
- In the "Conclusions" section, a justification should be included, as well as the main clinical lessons learned. Therefore, the last sentence should be deleted or removed to the "Discussion" section.
We have removed the last sentence and have rewritten the conclusion to provide justification and highlight lessons learned. Please see Lines 263-269.
- According to the CARE guideline, if possible, the patient's perspective or experience should be included.
Unfortunately, due to the retrospective nature of this report, we were unable to contact patients directly to get their perspective/experience. In its place, we attempted to include any documentation of the patient’s experience, such as reported side effects, that was provided in the patient’s medical chart.